# Herbicide Diuron as Endocrine Disrupting Chemicals (EDCs) through Histopathalogical Analysis in Gonads of Javanese Medaka (*Oryzias javanicus*, Bleeker 1854)

**DOI:** 10.3390/ani10030525

**Published:** 2020-03-20

**Authors:** Nur Amiera Kamarudin, Syaizwan Zahmir Zulkifli, Mohammad Noor Amal Azmai, Fatin Zahidah Abdul Aziz, Ahmad Ismail

**Affiliations:** 1Department of Biology, Faculty of Science, Universiti Putra Malaysia, UPM Serdang, Selangor 43400, Malaysia; nuramiera2510@gmail.com (N.A.K.); mnamal@upm.edu.my (M.N.A.A.); aismail@upm.edu.my (A.I.); 2International Institute of Aquaculture and Aquatic Sciences (i-AQUAS), Universiti Putra Malaysia, Batu 7, Jalan Kemang 6, Teluk Kemang, Si Rusa, Port Dickson 71050, Negeri Sembilan, Malaysia; 3Ministry of Energy, Science, Technology, Environment and Climate Change (MESTECC), Block C4 and C5, Federal Government Administrative Centre, Putrajaya 62662, Malaysia; zahidah_aziz@yahoo.com

**Keywords:** histology, ovary, testis, Diuron, Javanese medaka

## Abstract

**Simple Summary:**

The broadening of the agriculture domain around the world is causing excessive utilization of herbicides worldwide. The usage of herbicide Diuron towards the non-targeted aquatic organism may not directly cause death, but small sublethal doses of long-term exposure can cause detrimental consequences not only towards the individual itself but also towards the sustainability of the whole population of the species. The impaired reproduction of the gonadal staging and germ cells maturity can cause the overall reduction in adult survival and eventually lower the population density of the ecosystem.

**Abstract:**

The expeditious augmentation of the agriculture industry is leaving a significant negative impact on aquatic ecosystems. However, the awareness of the impacts of herbicide Diuron toxicities on the non-targeted aquatic organism, especially fish is still lacking. Javanese medaka, a new model fish species were exposed under sublethal levels and the long-term effects on gonads were investigated via histological studies. A total of 210 sexually mature fish were exposed to Diuron at seven different concentrations; control, solvent control, 1, 50, 100, 500, and 1000 μg/L for 21 days. In this study, Diuron caused histopathological alterations in gonads (ovary and testis) of Javanese medaka (*Oryzias javanicus*) by decreasing in gonadal staging and maturity of germ cells in oogenesis and spermatogenesis of female and male Javanese medaka. The results obtained in this study had proven our hypothesis that long-term exposure of herbicide Diuron can cause alterations in the gonadal histology of the adults of Javanese medaka.

## 1. Introduction

Diuron as a herbicide has been widely used in the field of agriculture as a total vegetation control. Diuron, with the chemical name of N-(3, 4-dichlorophenyl)-N, N-dimethyl urea (C_6_H_10_C_l2_N_2_O) belongs to the phenyl-urea family [1]. Herbicide Diuron is easily taken up by plants from soil solution through roots moving up via the xylem into the plant system of transpiration [2]. The total vegetation control is through the inhibition of the photosynthesis metabolic processes reaction by limiting the energy production compound of Adenosine Triphosphate (ATP) and preventing the production of oxygen. The inhibition mechanism involves blocking the electron transfer at the photosystem II complex level (Hill reaction) in the chloroplast thylakoid membranes [1,2].

A recent review of the literature on the contamination of Diuron around Peninsular Malaysia’s coastal waters has been conducted and it was reported that the expected sources of contamination were from the anthropogenic activities of antifouling paint coats from boats and ships [3]. Besides, the other factors that could possibly contribute to the contamination of Diuron are agriculture activities and the rainfall patterns in Malaysia. Some areas studied such as Johor, Klang, and Kemaman have a significantly high concentration of Diuron, suggesting that the sources of leaching and runoff from the terrestrial environment to the aquatic ecosystem are due to agricultural activities. Since herbicide containing Diuron is used as a vegetative control of grasses and weeds, it can eventually be found in the aquatic ecosystem, whereby it flows into the coastal water areas during the rain season. Besides, since Diuron is one of the main pesticides used in Malaysia, it has been hypothesized in a study [4] that the residue of herbicide Diuron can be extracted in crude palm oil (CPO) and crude palm kernel oil (CPKO) matrices. Another research was conducted by Muhammad et al. [5] on the leaching and persistence of herbicide Diuron in an oil palm plantation in Sepang, Selangor. The finding proved that Diuron is of moderate to high persistence in the soil, and this is influenced by the physico-chemical properties of the soil environment stability. The presence of Diuron residues can still be detected on the surface of the soil even after 90 d.

The status of Diuron contamination in Port Klang, Malaysia was updated in a recent study [6]. The study reported a significant amount of Diuron contamination in the sediment, pore-water, and surface seawater at Malaysia’s largest shipping port, Port Klang. The Diuron residues were highest in the concentration of sediment followed by porewater and surface seawater with the concentration range of 2.24 μg/kg to 19.28 μg/kg, 0.88 μg/L to 12.91 μg/L, and non-detectable to 0.53 μg/L. It was concluded that the sources of the Diuron contamination were from vessel mooring, rapid industrialization, active fishing activities, and rich flushing zones at the point sources in the Port Klang areas. Surprisingly, the recent concentration of Diuron residue in water samples had ranged up to 540 ng/L which surpassed the safe limit of Diuron (430 ng/L) [7] permitted to be applied on ships comparing to the previous report at the coastal area of Port Klang, Malaysia (285 ng/L) from the study by Ali et al. [3].

The Australian Pesticides and Veterinary Medicines Authority [8] stated a few steps to reduce the risk of Diuron in the environment by prohibiting antifouling paint containing Diuron from being applied on a boat less than 25 m long. This is to reduce the risks of presences of Diuron in the open sea and harbor by large vessels, ships or boats. Additionally, the waste from boat maintenance activities must be collected for all antifouling paint. A group of scientists from the United Kingdom, Lambert et al. [9] conducted a study on the assessment of the risk posed by antifouling biocide Irgarol 1051 and Diuron to freshwater macrophytes. Both these booster biocides were designed to inhibit the algal photosynthesis on the submerged surface of ships, boats and aquatic structures. The findings from this study reveal that the use of this booster biocide in the antifouling paint poses significant risks to *Apium nodiflurum* and *Chara vulgaris* in their growth rate, the maximum quantum efficiency of PSII, and root mass production.

Another research from the United Kingdom showed that antifouling paint booster biocide can be found in the coastal water as an input of antifouling directly from the painted hull and high-pressure hosing operations into the aquatic environment. Diuron can also be found in bottom sediments, as compared with other biocides such as clorothalonil, diclofluanid, and Sea-Nine 211 which are less persistent and rapidly removed from the water column [10]. Due to the banning of tributyltin (TBT) in antifouling paints in 2003, alternative booster biocides were used such as Irgarol and Diuron causing the concentrations of these toxic compounds to be high in the water column.

Japanese coastal waters have also been contaminated with antifouling compounds Irgarol and Diuron which were released from commercial antifouling paint. The research by Okamura [11], studied the 1011 photo-degradation of both these compounds after several months of exposure in different test waters under natural sunlight and kept in the dark as the control. The findings from this study concluded that under sunlight conditions, Diuron is more persistent than Irgarol as Irgarol underwent rapid degradation forming M1 (degradation product of Irgarol). The aquatic fate of these herbicidal compounds can ultimately impact the aquatic ecosystem in a longer time period.

Diuron toxicity had been studied in different parts of the world and most studies conducted by researchers in the developed countries had focused on using the Diuron biocide. The aquatic organism was the most studied in order to observe the effect of Diuron in the aquatic environment. According to a study [12], the histological analysis included a particular organ such as the reproductive organs, lung, digestive system, and foot tegument. Histological alteration and damages due to the exposure of Diuron was analyzed in comparison to the control and also based on previous experiences on the histology of the mollusc. There are several parts that could potentially be affected by Diuron due to direct contact; that is, the foot and mantel, the digestives system due to the enrolment in detoxification and metabolism, and the gonads related to the fertility of snails due to the exposure of Diuron. The alteration on the gonad can disrupt the endocrine activity [13]. The exposure of the biocide Diuron can also cause some effects on vertebrate’s behavioral changes [14].

Even though the purpose of the herbicide is to kill weeds occasionally, it has accidentally affected the aquatic ecosystems, particularly the non-targeted aquatic organisms such as fish. Therefore, this study was intended to investigate the effects of sublethal exposure of Diuron towards the native Malaysian fish species, Javanese medaka, which have the potential to be developed as specimens in the future. The gonads were analyzed since they are the prime organs of the fish population dynamics that determine the sustainability of aquatic ecosystems.

## 2. Materials and Methods

### 2.1. Collection and Maintenance of Javanese Medaka

Javanese medaka (*O. javanicus*) were the organisms tested in this study and they were collected at Sungai Pelek, Sepang, Selangor and subsequently, they were acclimatized in the flow-through freshwater system for 10 months until they produced new offspring in the Medaka’s laboratory (Department of Biology, Faculty of Science, Universiti Putra Malaysia). The maintenance protocols of Javanese medaka were derived from a recommendation of a previous study [15], whereby all the water parameters were closely monitored, the temperature, pH, dissolved oxygen and the mean results recorded (±SD) were 25 ± 1 °C, 7.0 ± 0.5, and 6.9 ± 0.6 mg/L, respectively [16]. The photo-period of the light/dark cycle was 14 h/10 h. Brine shrimps (*Artemia salina*) were used as the diet during the maintenance based on a 1–10% ratio of fish feed to body mass [17]. This is to ensure the survival and continuous reproduction of the Javanese medaka in the laboratory. The culturing of the brine shrimps was based on a previous study [18] where *A. salina* was used as the test organism.

### 2.2. Chronic Toxicity Testing

The chronic exposures of sublethal concentrations were conducted in 21 d in the semi-static water systems of 21 glass tanks. Each tank contained 6 l of test solution of different concentrations of Diuron and 10 individuals with the ratio 1:1 of sexually mature male and female Javanese medaka (age: approximately 12 m based on laboratory observation). The tanks with test solutions were continuously aerated and all the water parameters were monitored and checked regularly throughout the experiments. The treatment concentrations used were 1 µg/L, 50 µg/L, 100 µg/L, 500 µg/L, and 1000 µg/L; dechlorinated water controls, and; solvent controls (<0.03% of DMSO). The treatment concentrations were chosen from the probit analysis of acute exposure done at concentrations where all of the fish survived but may have significant impairment in the gonads. This sensitive early warning of a sublethal endpoint can be used to support regulatory pollutants assessments and monitoring to protect aquatic life [19]. The concentration of Diuron continued exposure was ensured by replacing the same concentrations of exposure in each tank once in three days due to the fact that the aqueous photolysis half-life of Diuron at pH 7 (25 °C) is around 43.1–2180 d [2]. The experimental protocol had met the Organisation for Economic Co-operation and Development (OECD) guidelines for the testing of chemicals on fish [20].

### 2.3. Ethics Statement

The fish were sampled, handled, treated and sacrificed according to the methods approved by the Institutional Animal Care and Use Committee, Universiti Putra Malaysia (AUP No.: R006/2016). All the experiments were conducted in accordance with the mentioned guidelines and regulations.

### 2.4. Histopathological Studies

After 21 d of sublethal exposure, Javanese medaka were sacrificed and fixed directly in Davidson’s solution for 24 h and transferred into a 10% buffered formalin for storage. The histopathological studies were perpetrated using the staining protocol of Haematoxylin and Eosin (R&M Marketing, Essex, UK), followed by the sectioning method of 5 µm thickness of a paraffin wax block. This method was also guided by [20] through the OECD guidelines for the fish histopathology gonad analysis. The histopathological changes in gonad sections were examined in randomly selected sections from the selected fish in each test solution. The histology slides were observed and pictured using Leica Histology Microscope (Leica Microsystems, Heerbrugg, Switzerland) with a color camera, which was connected directly to a monitor using a Leica’s Application Suite Version 3.2.0. For the analysis of the maturation of the ovary stages, the observed oocytes were divided into four stages: perinucleolar oocytes, cortical alveolar oocytes, late vitellogenic oocytes, and mature oocytes [21]. Atretic oocytes or atresia can be indicated by the breakdown of ovarian follicles and the irregular form of oocytes [20]. Meanwhile, testes were divided into five stages of seminiferous lobes containing germ cells: spermatogonia, primary spermatocytes, secondary spermatocytes, spermatids, and spermatozoa [20].

### 2.5. Statistical Analysis

Statistical analyses were performed using the statistical package of SPSS 22.0 at a 95 percent significance level. Differences among the treated group were analyzed using non-parametric Kruskal-Wallis followed by the Mann-Whitney test to compare differences between the groups. The significance of test results was ascertained at *p* < 0.05 [22].

## 3. Results

The development of Javanese medaka oocytes was divided into four stages based on the morphological features. The first stage was the primary growth of perinucleolar oocytes through the identification of the presence of small localized areas of intense basophilia in the cytoplasm. The second stage was the cortical alveolar oocytes which were identified through the appearance of cortical alveoli marks at the beginning of the formation of the vitellin envelope. The third stage was the late vitellogenic oocytes where their sizes were gradually increased due to the accumulation of the yolk. Finally, the last stage was the mature oocyte stage where the cytoplasm consisted of yolk instead of the nucleus as it had been dissolved. Figure 1a shows the micrograph of the controlled fish ovary in control treatments which exhibited the complete stages of an ovary in a female fish. In the first growth phase, the nucleus of the oocytes was filled with multiple nucleoli and the layer of the zona radiata was not thick in the growth phase.

Figure 1b shows the micrograph of the gonad section of ovarian at the 50 µg/L concentration with the presence of atretic oocytes, stromal haemorrhage, membrane retraction, and interstitial fibrosis. These adverse effects demonstrated the histoarchitectural changes in the fish ovaries due to the chemical exposure of Diuron which later had decreased the gonad growth. Additionally, based on Table 1, the developmental stages of oocytes are incomplete with only the presence of three stages at 500 µg/L of Diuron chronic exposure. Furthermore, Figure 2 shows the increases in the percentages of atresia (Figure 1c) formed during Diuron exposure.

Table 1 shows the histopathological analyses of the ovary based on the median gonadal staging present in the ovary and the percentages of atresia in the ovary that it was exposed to.

The complete seminiferous lobes containing germs cells should include the spermatogonia (SG), primary spermatocytes (SCI), secondary spermatocytes (SCII), spermatids (SD), and spermatozoa (SZ) to indicate that the fish is in a healthy state and can later release the spermatozoa for spawning activities. The testis of the controlled fish showed a typical organization of the testis with the presence of complete seminiferous lobes containing germ cells as shown in Figure 3a.

Figure 3b,c shows the disorganization of the lobules and disintegration of the Sertoli cells. The Sertoli cells were commonly occupying a peripheral location in the seminiferous lobules. The seminiferous lobes contain germ cells of spermatocytes, secondary spermatocytes, and spermatids at 50 µg/L concentration. The seminiferous lobules were held together by the connective tissues and muscles fibres. The same process was observed at 100 µg/L concentration (Table 2) with only the presence of three medians of gonadal staging.

The median numbers of gonadal staging are summarised in Table 2 after the Diuron was exposed for 21 d at different concentrations. The control was at the lowest Diuron exposure of 1 µg/L and the median gonadal staging was in the complete phase. Meanwhile, at the concentration of 50 µg/L and 100 µg/L, the number of median staging was decreased to three stages, that is, spermatocytes (SCI), secondary spermatocytes (SCII), and spermatids (SD). At the concentration of 500 µg/L and 1000 µg/L, the median stage was the only one where the spermatids compacted in the seminiferous lobules.

The numbers of median gonadal staging were summarized in Table 2 after the 21 d exposure of Diuron at different concentrations. The control and at the lowest of Diuron exposure of 1 µg/L, the median gonadal staging for thesis is in the complete phase, While, at the concentration of 50 µg/L and 100 µg/L, the number of median staging decreases to three stages of spermatocytes (SCI), secondary spermatocytes (SCII), spermatids (SD) at the concentration of 500 µg/L and 1000 µg/L, the median stages is the only one in which the spermatids compacted in the seminiferous lobules.

## 4. Discussion

Histological alteration was observed in terms of reproductive organs or gonads in order to study the effect of sublethal concentrations of herbicide Diuron towards the population of fish through alteration in the gonads (ovary and testis). The results obtained in this study had proven our hypothesis that long term exposure of Diuron can cause alterations in the gonadal histology of adult Javanese medaka. The normal ovary structure showed the complete four gonadal follicular stages, namely perinucleolar oocytes, cortical alveolar oocytes, late vitellogenic oocytes, and mature oocytes in both control and solvent-control fish at the lowest Diuron exposure of 1 µg/L. The physical structure of oocytes did not show any obvious alteration in both control treatments. However, when the concentration of Diuron was increased, the median stage of gonadal staging decreased and changed the oocytes’ structure.

In this study, we can conclude that the contamination of herbicide Diuron in the environment can lead to a reduction in the number of oocytes which can cause reduced fecundity of the female fish. According to a study by Johnson et al. [20], the reproduction performance of oocyte can be identified through the increasing number of atretic oocytes present in the ovary. The atresia of oocytes can be initiated by metabolic factors, endocrinological, and any disturbance in the environment [23]. In addition, according to Agarwal et al. [24], there are several other factors that can cause atresia such as hypophysectomy, steroids, captivity, starvation, and biocides.

These estrogenic activities of Diuron and its metabolites were recently studied by Pereira et al. [25] and proved that the exposure of Diuron can interfere with the normal functions of the endocrine system of fish as one of the Endocrines Disrupting Chemicals (EDCs). EDCs are a class of environmental pollutants that alter the estrogenic functions of the organism through the disruption of the estrogen receptors [26]. Additionally, the major mechanisms of endocrine disruption mentioned by Carnevali et al. [27] are the EDC binding to the hormone receptor as well as the inhibition or stimulation of hormone metabolism. The findings of this study were also in parallel with a study by Pereira et al. [25] which also observed the ovary of a model fish called *Oreochromins niloticus* or Nile tilapia at the exposure of 0.1 µg/L of Diuron. The examination of the ovarian lamellae morphometry showed a decrease in the percentages of the primary ovarian follicles, an irregular shape of oocytes, and a large number of cortical alveoli vesicles present. The well-known EDCs, Dichlorodiphenyltrichloroethane, or also commercially known as Dichlorodiphenyltrichloroethane (DDT) showed poor gonadal development in the fish collected from the DDT-affected areas. The study by McHugh et al. [28] showed that the presence of gonadal staging of the female specimens was only pre-vitellogenic oocytes, and the ovaries were filled up with adipose tissues within the visceral cavity. Furthermore, Diuron and its biodegradation metabolites act as endocrine disruptors which have subsequent effects on the reproductive development of fish.

Another phenyl-urea herbicide atrazine showed a reduction in the reproduction of Japanese medaka (*O. latipes*) on their total eggs produced in all atrazine-exposed groups (0.5, 5.0, and 50 µg/L) as compared to the control group [29]. This is related to the decrease in the number of eggs ovulated and oocyte development and production. The study by Papaolias et al. [29] suggested that exposure to atrazine can alter the final maturation of oocytes which is related to the reduction in egg production. Similar findings were also reported by Boscolo et al. [30] and verified that this herbicide can reduce egg production in fish which attributed to the reduction in spawning events. They had also observed abnormalities in the lipid accumulation and atretic follicles in the exposed fish at an atrazine concentration of 0.5, 5.0, and 50 µg/L. According to a study by Tillitt et al. [31], the acute toxicity of Diuron appears to be similar to linuron in fish. Therefore, the gonadal histology of the linuron exposed fish fathead minnow (*Pimephales promelas)* also showed the under-development of oocytes in which a majority of the observed follicles were in mid-to-late vitellogenic oocytes in all treatments (1, 10, 100 μg/L) [32].

There are limited studies that assessed the impacts of common herbicides on fish reproduction that can actually cause potential reproductive and endocrine damage. In the current study, Diuron had caused changes in the testis by inducing the incomplete seminiferous lobes containing germs cells and had decreased the gonadal staging scale as the concentration of exposure became higher. The complete seminiferous lobes containing germs cells should include the spermatogonia (SG), primary spermatocytes (SCI), secondary spermatocytes (SCII), spermatids (SD), and spermatozoa (SZ) to indicate that the fish is fit for reproduction and can later release the spermatozoa for spawning activities. In addition, the stages of spermatogenesis progress such as the undeveloped stage, early spermatogenic stage, mid-spermatogenic stage, late spermatogenic stage, and spent stage were used as the recommended indicators for male Javanese medaka gonadal staging.

The results of the present study clearly demonstrate the degenerative changes in testicular lobules. In the present study, the exposure of Diuron to Javanese medaka had also caused testicular damage such as the disorganization of the lobules and disintegration of the Sertoli cells. This condition could lead to a reduction in the number of spermatozoa. The gonadal staging in the testis of Javanese medaka also showed a decrease in the number of stages in spermatogenesis. These results obtained are crucial in verifying that the exposure of Diuron, even at a low concentration can cause significant damage to the reproductive functions and gonadal development and growth.

In another study by Bauer et al. [33], the findings showed that the exposure of Diuron can affect estrogenic activities in Nile tilapia as the testosterone levels were decreased by Diuron exposure. This study concluded that the herbicide and its metabolites can potentially cause reproductive impairments in male fish since they have the anti-androgenic activity which is also known as testosterone blockers. They have also proven that the exposure of Diuron metabolites which are more toxic had caused a decrease in the percentages of seminiferous tubules and mean percentages of spermatids, spermatozoa and the number of Sertoli cells per cross-section of the seminiferous tubules. These same effects could exert at higher concentration exposure of Diuron towards the smaller model organism used in this study.

In a study by Felício et al. [34], it was observed that Diuron has shown to antagonize estrogenic and androgenic receptors of steroid hormones which can potentially impair the gonadal steroidogenesis and the reproductive system of teleosts. Diuron has also shown to be anti-androgenic, inducing vitellogenin in juvenile male tilapia (*Oreochromis mossambica*) [35] and anti-androgenic compounds (AAC) with key enzymatic activities [36]. Another effective herbicide called Atrazine also demonstrated the reduction in the reproduction of Japanese medaka (*O. latipes*) [29] and fathead minnow (*Pimephales promales*) [30]. Both studies showed no significant difference in the mean Gonado-Somatic Index (GSI) values. However, testicular staging showed significant changes with an increased number of abnormal mitoses in spermatogonial cells by two to three times the number for unexposed fish. The effects of Diuron exposure were also previously studied on the spermatogenesis of the male lizard *Podarcis sicula* [13]. The results obtained from this study showed a complete loss of meiotic and mature germ cells; reduction of primary spermatocytes, secondary spermatocytes, and spermatids; and a decrease of spermatozoa in the treatment groups.

## 5. Conclusions

Based on the observations in this study, long term exposure of Diuron may attribute to the reduction in the oocyte number which may also reduce the spawning activities of fish population. The changes in the ovarian tissue structure and reduction in the presence of gonadal staging may also lead to fewer viable eggs and disruption of the teleost oogenesis, and finally affect the population dynamics of the fish in the polluted aquatic ecosystem. Diuron can also cause changes in the testis by inducing the incomplete seminiferous lobes containing germs cells and decrease in the gonadal staging scale as the concentration of exposure increases. These results obtained were a crucial indicator that demonstrated the sublethal concentration of Diuron exposure may cause severe damage, resulting in the histological alterations of reproductive organ dysfunction as the alterations started showing even at the lowest concentration of 1 µg/L Diuron exposure. The organs of Javanese medaka that were observed were also a good sample to demonstrate Diuron toxicity in the freshwater ecosystems and even for marine ecosystems due to its uniqueness. Our findings call for a future study on the adverse effects on reproductive impairments from the exposure of Diuron metabolites, DCA which happen to be more toxic in the aquatic ecosystem.

## Figures and Tables

**Figure 1 animals-10-00525-f001:**
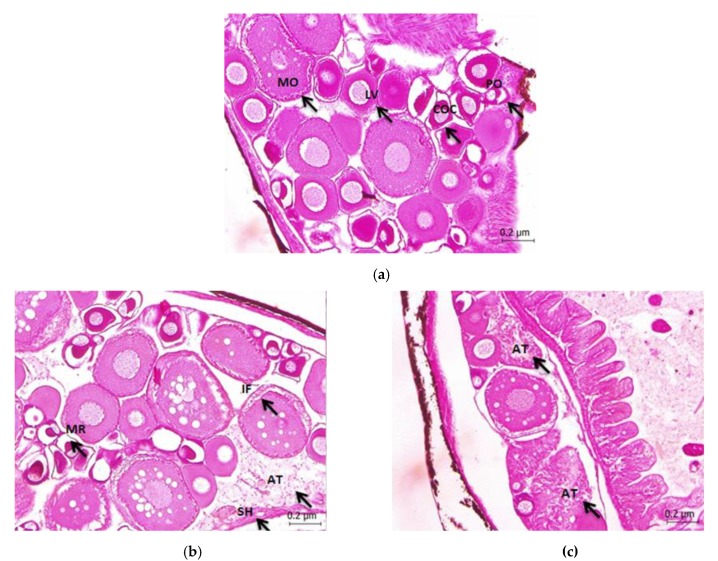
(**a**) Micrographs of dechlorinated water control treatment of Javanese medaka’s ovary section (5 µm) stained with H and E × 400. PO-Perinucleolar Oocytes, COC-Cortical Alveolar Oocytes, LV- Late Vitellogenic Oocytes, MO- Mature Oocytes. (**b**) Micrographs of 50 µg/L of Diuron chronic exposure of Javanese medaka’s gonad section (ovary) (5 µm) stained with H and E × 400. Stromal hemorrhage (SH), atretic follicle visible within an ovary (AT), membrane retraction (MR), and Interstitial Fibrosis (IF). (**c**) Micrographs of 500 µg/L of Diuron chronic exposure of Javanese medaka’s gonad section (ovary) (5 µm) stained with H and E × 400. The ovary shows the increasing of the atretic follicle and decreasing of the follicular stage LV: Late Vitellogenic Oocytes; MO: Mature Oocytes.

**Figure 2 animals-10-00525-f002:**
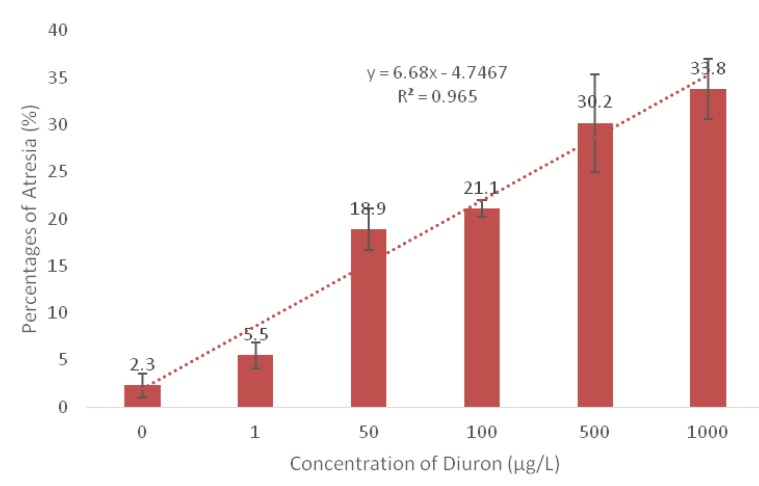
The percentages of atresia of oocytes increases with the trend line of R^2^ = 0.965 at dechlorinated water control (0 µg/L), 1 µg/L,50 µg/L,100 µg/L,500 µg/L, 1000 µg/L of Diuron exposure.

**Figure 3 animals-10-00525-f003:**
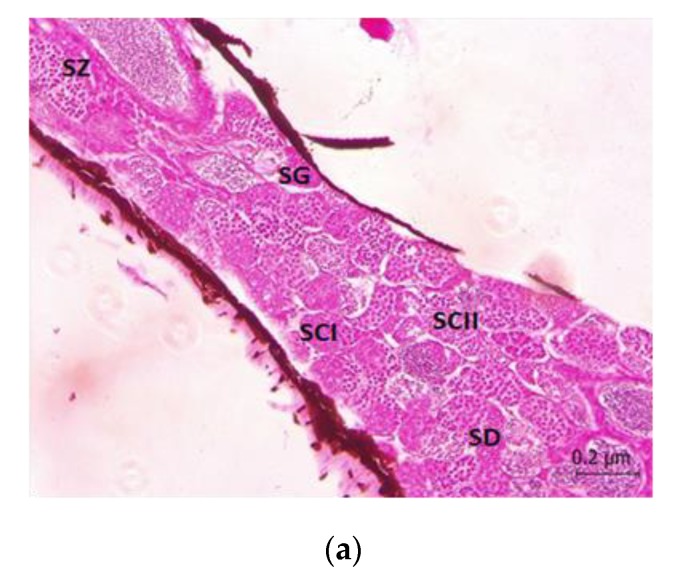
(**a**) Micrographs of dechlorinated water control treatment of Javanese medaka’s testis section (5 µm) stained with H and E × 400. Seminiferous lobes containing germ cells: spermatogonia (SG), primary spermatocytes (SCI), secondary spermatocytes (SCII), spermatids (SD), spermatozoa (SZ). (**b**) Micrographs of 50 µg/L of Diuron chronic exposure of Javanese medaka’s gonad section (testis) (5 µm) stained with H and E × 400. Disorganization of the lobules and disintegration of the Sertoli cells. Seminiferous lobes containing germ cells spermatocytes (SCI), secondary spermatocytes (SCII), spermatids (SD). (**c**) Micrographs of 1000 µg/L of Diuron chronic exposure of Javanese medaka’s gonad section (testis) (5 µm) stained with H and E × 400. The seminiferous lobes contain spermatids (SD). The seminiferous lobes contain spermatids (SD). Some part of testis cells are undergoing necrosis.

**Table 1 animals-10-00525-t001:** Histopathologic analyses of the ovary of Javanese medaka (*O. javanicus*) exposed to sublethal Diuron during 3 w with the percentages of atresia and median gonadal staging (*n* = 36).

Concentration Treated (µg/L)	Increases in Oocyte Atresia (%)	Gonadal Staging Median Stage
Control	2.0	4 ^a^
1	5.5	4 ^a^
50	18.9	3 ^b^
100	21.1	2 ^ab^
500	30.2	2 ^ab^
1000	33.8	2 ^ab^

^ab^—indicates a significant difference (*p* < 0.05) between the groups.

**Table 2 animals-10-00525-t002:** Histopathological analyses of testis of Javanese medaka (*O. javanicus*) exposed to sublethal Diuron during 3 weeks with the median gonadal staging (*n* = 36).

Concentration Treated (µg/L)	Gonadal Staging Median Stage
Control	5 ^a^
1	5 ^a^
50	3 ^b^
100	3 ^b^
500	1 ^c^
1000	1 ^c^

^abc^—indicates a significant difference (*p* < 0.05) between the groups.

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
