# Peer review of "Herbicide Diuron as Endocrine Disrupting Chemicals (EDCs) through Histopathalogical Analysis in Gonads of Javanese Medaka (Oryzias javanicus, Bleeker 1854)"

_animals, 2020, doi:10.3390/ani10030525_

Round 1

Reviewer 1 Report

The manuscript by Nur Amiera Kamarudin et al. describes the effect of herbicide Diuron on ovary and testis of Javanese medaka. The effect of Diuron was investigated by histological analysis. The topic is very relevant and the data presented novel.

I would recommend this article for publication after major revision.

I would like to draw your attention mainly to chemical analyses of water and confirmation of exposure concentration of Diuron.

General and Specific comments:

Simple summary. It is nice introduction, but it not show any outcome from present work. Diuron was not mentioned, as well as its effect on medaka gonads after chronic exposure.

Abstract. Line 32-34. Is it results of present study or another, recently published? It is confusing sentence.

Introduction. Provide environmentally relevant concentration of Diuron in water. Justify of selection of exposure concentrations.

MM

Line 96. Specify all water parameters, which were investigated.

Line 100. Specify % of fish feed which fish ate in one day from body mass.

How actual exposure concentration was confirmed? Provid ethis information. Add information of Diurone concentration stability in water.  How many times water was changed to support continues exposure with constant concentration of Diuron? 

Line 112. Avoid "killed". Slaughtered or sacrificed. What mortality was observed during the test? In conclusion it is mention, that Diurone  can result in death.

Line 110 and 116. Add number of used OECDs.

Provide numbers of regulation/license on conduction such experiment and Animal Care Advisory Committee.

Provide a table of semiquantitative evaluation for each fish (can be supplementary).

The manuscript is missing description of statistical analyses, which was performed for data in table 1 and 2.

Figure 2. Table 1, 2. Provide number of fish investigated.

Conclusions. Specify  Diurone concentration, which made histological alteration in fish gonads.

Line 295. It is strong conclusion, that Diuron can result in death. Provide evidence on fish mortality during the test.

Reviewer 2 Report

This manuscript contains important and interesting data of diuron effcts on gonads in Javanese medaka.

Major comments:

Lines 30, 140: “sexually mature” should be replaced by the “age” of fish used. Lines 70-72: Authors described that the safe limit of diuron, 430 ng/L. Need to add a reference indicating this value. Also need to explain the reason why the safe limits are different between 430 and 285 ng in the reference [3]. Procedures to decide the safety limit should be explained. Lines 143, 152, 180: authors used “control”, however, need to describe, control with solvent or without solvent should be clarified, since <3% DMSO is unusual. Lines 222-235: authors need to add concentrations of chemicals used in each literature, otherwise, readers will not understand the concentrations are extremely high or environmentally relevant. This is very important to understand chemical effects. Line225: what is the meaning of “disruption of the estrogen receptors”? Receptor protein damage? Lines 236-239: need to add reference here. Reference 18 was not cited in this manuscript. Lines 239-246: use concentrations of chemicals for explanation and discussion. Lines 259, 270: sertoli cells should be Sertoli cells. Sertoli is a scientist name who found this cell. Line 265: Present reviewer cannot understand this line. Need to rewording. Lines 273-274: “estrogenic and androgenic receptors” What is this mean? Estrogenic effect of diuron is discussed in this manuscript, however, levels of estrogenic activity has not been mentioned. Better to find the estrogenicity of diruon should be used in discussion. Estrogenic potency of 17b-estradiol as 100%, then diuron is ?%of the estradiol? In vitro data will be easily found maybe. Over all, authors used quite high concentrations, compared to the environmentally detected concentration, to induce changes in gonadal histopathology. Therefore, authors need to evaluate the potential risk of diuron concentration at present in the discussion and conclusion.

Minor comments

Line 344; what the “” of the title of the reference 13? Line 349: book title should be large capital?

Reviewer 3 Report

The manuscript “Herbicide Diuron as Endocrine Disrupting Chemicals 2 (EDCs) through histopathalogical analysis in gonads 3 of Javanese medaka (Oryzias javanicus, Bleeker 1854)” reports the study on the impacts of Diuron exposure on Javanese medaka. Diuron is a widely used herbicide and could pose potential damage for aquatic organisms. It is important to understand how Diuron could affect aquatic organisms especially fish.  Here are my specific comments.

The author should pay attention to the way the references were cited in the paper. Please refer to the author’s guidance. For example, Line 46, the reference 1 and 2 should be cited like “chloroplast thylakoid membranes [1,2].” Also, reference [3] and ref [15] were not cited properly. There is very limited literature in the introduction session on the toxicity of Diuron. The author should also talk about the potential mechanisms. Is there any regulation or control on Diuron? It would be good if the author can include that in the introduction session. Methods session, how many male and female Javanese medaka were used in the study? Is the sample size enough to support the conclusion? The author should describe the details of the experiment. How is the percentage of Atresia calculated in Figure 2. Please provide the details of the method in the method session. Figure 2, caption should be rephrased. Why the figure for control sample in Table 1 is not consistent with the figure in Figure 2. Some paragraphs in discussion session should be moved to the introduction part. The results from this study are not fully discussed like the impact of the finding, the mechanism behind the results, the limitation of the study and future plan. Author contribution: Please provide the right name of the authors and their contribution to the manuscript. Acknowledgement session, it seems the author copy and paste from the guidance. The authors can delete this session if they have no support to acknowledge. Grammar need to be carefully checked before publication.

Round 2

Reviewer 1 Report

Dear Editor,

Please find below my comments on Manuscript Number: animals-710338  entitled „Herbicide Diuron as Endocrine Disrupting Chemicals (EDCs) through histopathalogical analysis in gonads of Javanese medaka (Oryzias javanicus, Bleeker 1854)”.

Authors replied almost on my all major concerns. The manuscript is missing important information of actual Diuron exposed concentration.

In the manuscript is mentioned:

“The concentration of Diuron continues exposure were ensure by replacing the same concentrations of  exposure in each tanks once in three days”.

Does Diuron exposed concentration was stable for 3 days? Does aeration had effect on Diurone concentration? This information is crucial. It is not enough put 1 mg of substance to 1 L water and expect that it will be 1 mg/L.

Another comment is related to description of statistical analyses section. What significance level was chosen? Does data were checked on normality?

Line 33. Delete - recently reported to have

Reviewer 3 Report

The manuscript “Herbicide Diuron as Endocrine Disrupting Chemicals 2 (EDCs) through histopathalogical analysis in gonads 3 of Javanese medaka (Oryzias javanicus, Bleeker 1854)” is greatly improved after the revision. There are some things need to be fixed before publication. 

The authors still need to pay attention to the way the reference cited in the paper. For example, Line 74,75,79,272, 287, the authors need to refer to the cited reference (e.g. authors name of the cited reference) instead of just using the reference number.  Grammar still need to be checked before publication. There are still some sentences that doesn’t read right.
